# Structured-Light 3D Imaging Based on Vector Iterative Fourier Transform Algorithm

**DOI:** 10.3390/nano14110929

**Published:** 2024-05-25

**Authors:** Runzhe Zhang, Siyuan Qiao, Yixiong Luo, Yinghui Guo, Xiaoyin Li, Qi Zhang, Yulong Fan, Zeyu Zhao, Xiangang Luo

**Affiliations:** 1National Key Laboratory of Optical Field Manipulation Science and Technology, Chinese Academy of Sciences, Chengdu 610209, China; kuriharaaiko@foxmail.com (R.Z.); qiaosiyuan22@mails.ucas.ac.cn (S.Q.); luoyixiong21@mails.ucas.ac.cn (Y.L.); gyh@ioe.ac.cn (Y.G.); lixiaoyin@ioe.ac.cn (X.L.); zhangqi@ioe.ac.cn (Q.Z.); fanyl@ioe.ac.cn (Y.F.); 2State Key Laboratory of Optical Technologies on Nano-Fabrication and Micro-Engineering, Institute of Optics and Electronics, Chinese Academy of Sciences, Chengdu 610209, China; 3Research Center on Vector Optical Fields, Institute of Optics and Electronics, Chinese Academy of Sciences, Chengdu 610209, China; 4School of Optoelectronics, University of Chinese Academy of Sciences, Beijing 100049, China; 5Tianfu Xinglong Lake Laboratory, Chengdu 610299, China

**Keywords:** quasi-continuous-phase metasurface, diffractive optical elements, iterative Fourier transform algorithm, structured-light 3D imaging

## Abstract

Quasi-continuous-phase metasurfaces overcome the side effects imposed by high-order diffraction on imaging and can impart optical parameters such as amplitude, phase, polarization, and frequency to incident light at sub-wavelength scales with high efficiency. Structured-light three-dimensional (3D) imaging is a hot topic in the field of 3D imaging because of its advantages of low computation cost, high imaging accuracy, fast imaging speed, and cost-effectiveness. Structured-light 3D imaging requires uniform diffractive optical elements (DOEs), which could be realized by quasi-continuous-phase metasurfaces. In this paper, we design a quasi-continuous-phase metasurface beam splitter through a vector iterative Fourier transform algorithm and utilize this device to realize structured-light 3D imaging of a target object with subsequent target reconstruction. A structured-light 3D imaging system is then experimentally implemented by combining the fabricated quasi-continuous-phase metasurface illuminated by the vertical-cavity surface-emitting laser and a binocular recognition system, which eventually provides a new technological path for the 3D imaging field.

## 1. Introduction

Light, as a kind of electromagnetic wave, is an important carrier for transmitting energy and information. The control of light is an important and popular topic in the field of optics. Reflection and refraction are almost the earliest means of light control used by human beings, which mainly involve the use of mirrors and lenses. The optical resolution of these devices is mainly dictated by the diffraction limit. Recently, along with the advancement of CMOS-compatible nanofabrication technology, the metasurfaces obeying the generalized Snell’s law can manipulate light’s refraction and reflection with high efficiency by eliminating the high-order diffraction [1]. Metasurfaces allow tailoring the parameters of amplitude, phase, and polarization of light at a sub-wavelength scale beyond the capability of natural materials [2,3,4]. Metasurfaces show a wide range of potential applications in the fields of imaging [5,6,7], photon recognition [8], electromagnetic manipulation [9,10,11], optical manipulation [12,13,14], optical encryption [15], neural networks [16,17], measurement [18,19], and detection [20]. Catenary metasurfaces, as a kind of high-quality metasurface with broadband high conversion efficiency endowed with continuous phase modulation, are widely used [21].

Compared to conventional discrete-phase metasurfaces, quasi-continuous-phase metasurfaces consist of trapezoidal dielectric or metal pieces that exhibit better performances [22,23,24]. In particular, catenary optics solves the high efficiency and broadband phase modulation that can be hardly achieved by discrete microstructures [21]. Diffractive optical elements (DOEs) can control the spatial distribution of a beam using a single element, which has the advantages of high flexibility, small size, and high efficiency. DOEs show great realistic application potential in the fields of optical imaging [25], nanoscale printing [26], medical research [27], structured light illumination [28,29,30], and industrial use [31]. With the development of structured-light face recognition technology on smartphones, wide-angle laser beam-splitting DOEs are needed [32,33,34]. Herein, the maximum diffraction angle and uniformity error are two important indexes to determine the performance of the beam splitter. A traditional DOE design method based on scalar diffraction theory makes it difficult to achieve large diffraction angles and always suffers from low uniformity due to the limitations of paraxial approximation and electromagnetic coupling [35].

To achieve accurate design optimization of large-angle laser beam-splitting devices, an accurate evaluation of the beam-splitting effect of the devices with the help of vector electromagnetic simulation is necessary. To this end, we choose an optimization algorithm based on vector diffraction theory that can balance the computation cost and design efficiency, which is referred to as the vector iterative Fourier transform algorithm (IFTA) in this work.

Structured-light three-dimensional (3D) imaging is a popular non-contact 3D topography measurement technology [36], which has the advantages of low hardware requirement, low computation, high imaging accuracy [37], high dot density, fast imaging speed, and low cost. This technology finds potential applications in the field of 3D imaging [38]. In structured-light 3D imaging, how to design DOEs to generate uniform point light sources is the key technical bottleneck. The combination of quasi-continuous-phase metasurface and IFTA can effectively improve the uniformity of the dot matrix light source, thus achieving 3D imaging and target reconstruction with higher quality.

In this work, we introduce IFTA and design a 7 × 7 DOE based on this algorithm with a maximum diffraction angle of 35° at 940 nm, achieving an overall diffraction efficiency of 79%. Combined with a vertical-cavity surface-emitting laser (VCSEL), we achieve structured-light 3D imaging and target reconstruction with a field of view up to 35° × 35° and an effective number of 8869 projected points. It provides a new technical path to practical application for 3D imaging technology.

## 2. Principles and Methods

### 2.1. Design of 7 × 7 DOE with Uniform Energy Distribution

The well-known GS algorithm proposed by Gerchberg and Saxton is the most used in scalar diffraction theory and is also known as the iterative transform Fourier algorithm [39]. It is capable of recovering the phase distribution in the diffraction plane under the precondition that the field strength distributions in the image and diffraction planes are known. The drawbacks of the GS algorithm are that it tends to fall into a local optimum solution and that the target image also influences the iterative process. When only using the GS algorithm, the design of beam splitter DOEs with large diffraction angles makes the spot intensity non-uniform, which can be an influencing factor. Here, we add certain constraints to the GS algorithm as the initial solution of the whole algorithm.

Figure 1 illustrates the optimization method for Vector IFTA, and the specific flow of the algorithm is described in detail below. In the practical operation of the iterative Fourier transform algorithm, the target image will generally add zero. This can appropriately release the constraints in these regions. The singularity can be better suppressed which can improve the imaging quality. So, we carry out the following processing:(1)$$C_{K\_image}=\frac{\left|G_{K\_normalized}\right|}{\left[A_{image}+1/\infty \right]}$$
(2)$$w_{K\_image}=w_{K-1\_image}\times C_{K\_normalized\_image}$$
where “*image*” means the part without adding zero. *G* represents the complex amplitude in the frequency domain and *K* represents the number of steps iterated in the algorithm. $A_{image}$ represents the amplitude of the target image in the output plane; 1/∞ is a sufficiently small number. The initial value of *w* is 1.
(3)$${G^{\prime }}_{K\_all}=A_{all}\cdot w_{K\_all}\cdot \exp[i\phi (G_{K\_all})]$$
where the subscript “*all*” represents all the regions including the complementary zeros, and $\phi (G_{K\_all})$ signifies the phase of the complex part. To suppress the singularity, the complex amplitude in the frequency domain needs to be corrected as follows:(4)$$c=b\cdot \sum \left|{G^{\prime }}_{K\_image}\right|/L$$
(5)$${G^{\prime }}_{K\_image}(|{G^{\prime }}_{K\_image}|>c)=c$$

The value *b* is an artificial parameter whose value affects the speed of convergence of the algorithm. *L* is the number of elements in the target region. After the above operation, the adjusted complex amplitude in the frequency domain is obtained, and the above steps are repeated in the iteration between the output plane and the input plane, which can effectively improve image quality.

After obtaining the initial solution by the revised GS algorithm, we obtain the binary phase for the beam splitter.
(6)$$\phi (x,y)=\left\{\begin{array}{cc} \pi /2 & if\varphi [g(x,y)]\geq 0\\ -\pi /2 & if\varphi [g(x,y)]\leq 0 \end{array}\right.$$

g(x,y) represents the complex amplitude in the time domain, and φ[g(x,y)] is the function to obtain the phase information. Once the binary phase distribution is obtained the beam splitter device can be modeled. The structural material fills into the designed region of the device and the unfilled portion of the space is air.

After determining the period and height of the beam splitter unit according to the grating equations, a vector electromagnetic simulation can be performed to obtain the diffraction efficiency $\eta_{K}$, which is optimized according to the diffraction intensities of each part of the beam splitter, and the following evaluation function *EF* is obtained.
(7)$$EF=\frac{\eta_{K\max}-\eta_{K\min}}{\eta_{K\max}+\eta_{K\min}}$$
where $\eta_{K\max}$ and $\eta_{K\min}$ refer to the maximum and minimum diffraction efficiency, respectively. During the whole iteration, if *EF* reaches the optimization condition or the number of iterations reaches the upper limit, the optimization stops. During one generation of iteration, if *EF* is better than before, the amplitude value of this *EF* will be saved, or else the algorithm will continue.

To progressively arrive at a uniform distribution of energy in each unit of the beam splitter, we updated the amplitude to reduce the energy of the strongest unit and increase the energy of the weakest one, with the correction function $M(A_{K},\eta_{K})$ defined as follows:(8)$$A_{K+1\_\max/\min}=M(A_{K},\eta_{K})=\frac{A_{K\_\max/\min}}{{\left(\eta_{K\_\max/\min}/\eta_{K\_average}\right)}^{m}}$$
where $A_{K\_\max/\min}$ is the amplitude corresponding to the diffraction order with the *K*th largest or smallest diffraction efficiency. $\eta_{K\_average}$ is the average value of $\eta_{K}$. $\eta_{K\_\max/\min}$ is the strongest or weakest diffraction efficiency and *m* is the interference coefficient, which is used to further optimize when the algorithm starts to converge. *m* varies between 0.7 and 0.9 freely in the simulations below.

### 2.2. Point Matching in Structured-Light 3D Imaging and Target Reconstruction

The point matching technique is to find the mapping relationship between different point clouds under different viewpoints. It utilizes specific algorithms to convert the point clouds of the same target to a unified coordinate system. The process of point cloud matching is the process of matrix change. In structured-light 3D imaging, the points extracted from the left and right camera images captured by the binocular vision system are matched to the corresponding points by the point cloud matching technique, which provides the necessary data support for the subsequent 3D reconstruction.

The Iterative Closest Point (ICP) algorithm, proposed by Besl et al. in 1992, is a classic algorithm in the field of 3D point matching [40]. The algorithm uses distance as the optimization objective. It iterates to estimate the rigid-body transformation between the source and target point clouds and then transforms the source point cloud. It also calculates the transformed mean square deviation to determine whether the iteration is terminated or not. The ICP algorithm is intuitive, understandable, and implementable. But, the requirements for the initial positions of the two groups of point clouds are stringent; otherwise, it is easy to fall into the local optimum, which affects the matching accuracy.

In 2003, Chui et al. proposed the Robust Point Matching (RPM) algorithm [41], which uses annealing and soft correspondence to assign a value between 0 and 1 to any pair of points, and eventually converges to 0 or 1. If the final value is 1, it means that the two points are a matched pair. The RPM algorithm eventually computes a one-to-one mapping of the matched pairs, whereas the ICP algorithm usually does not. However, the RPM algorithm is likely to fail when there are noisy points or some structures are missing.

The Kernel correlation (KC) algorithm [42], proposed by Tsin et al. in 2004, is a similarity measurement. It reduces the distance between left-camera and right-camera point clouds by measuring their similarity. The value of the global objective function will gradually decrease and converge when the KC algorithm is optimized. The complexity of the KC algorithm is high because it requires comparing the points one by one. Compared to others, it takes a long time to compute.

In 2010, Myronenko et al. proposed the Coherent Point Drift (CPD) algorithm [43], which converts the points matching problem into a probability density estimation problem by representing the distribution of the source point clouds as a mixed Gaussian model. The corresponding likelihood function reaches the maximum when the source point clouds and the target point clouds are matched. The CPD algorithm is a very robust algorithm, which is not sensitive to the position and gestures of the initial point clouds. It can also deal with noise and local deformation. Moreover, the CPD algorithm does not need a priori information, such as the exact position of the initial corresponding points. It can automatically recognize the correspondence by the coherence between the point clouds, which is suitable for point clouds with different shapes and densities. Because of the above advantages of the CPD algorithm, we use it to realize point matching in reconstruction.

In the CPD algorithm, the alignment of the left-camera point cloud and right-camera cloud is modeled as a probability density estimation problem. One point cloud is represented by a Gaussian Mixture Model (GMM) and the other point cloud matches the first one by moving as a whole. We define the point clouds of the two cameras in the binocular vision system are $X=(x_{1},x_{2},\ldots ,x_{U})$ and $Y=(y_{1},y_{2},\ldots ,y_{V})$, respectively, where *U* and *V* represent the total number of the point clouds of the two cameras, separately. The transformation relation between *X* and *Y* is X=T(Y,θ), where *θ* is the parameter that defines *T*, such as the rotation matrix, translation vector, and scaling factor. We set *Y* as the center of the GMM, and the point GMM probability function is expressed as follows:(9)$$\left\{\begin{array}{l} p(x)=\sum_{u=1}^{U+1}P(u)p(x|u)\\ p(x|u)=\frac{1}{2\pi \sigma^{2}}e^{\frac{|x-y_{u}{|}^{2}}{2\sigma^{2}}} \end{array}\right.$$

For all GMM members, P(u)=1/U. Considering the presence of noise in the data, U≠V, the following additional uniform distribution can be introduced in this case.
(10)$$p(x|U+1)=\frac{1}{V}$$

Combining Equations (9) and (10), weights factor *q* are introduced, where 0≤q≤1. The complete probability density function after weighting the above two formulas is the following:(11)$$p(x)=q\frac{1}{V}+(1-q)\sum_{u=1}^{U}\frac{1}{U}p(x|u)$$

Assuming that each data point is independently and identically distributed, the likelihood function can then be written as follows:(12)$$E(\theta ,\sigma^{2})=\sum_{u=1}^{U+1}\log\sum_{v=1}^{V+1}P(u)p(x|u)$$

The desired parameters $\left(\theta ,\sigma^{2}\right)$ are obtained by maximizing the above likelihood function. Typically, solving the problem of maximizing the likelihood function involves taking the logarithm of the likelihood function, solving for the derivative, and constructing a system of equations by setting the derivatives to zero in order to obtain the desired parameters. However, due to the complexity of the equations here, solving for the parameters using the above methods can be cumbersome. The Expectation Maximization (EM) [44,45] algorithm could be a good way to obtain parameters. The EM algorithm is an iterative method for finding the maximum likelihood estimates of the parameters in a statistical model. It consists of two steps: the expectation step calculates the expected value of the likelihood function based on the current estimates of the parameters, and the maximization step updates the parameters by maximizing the expected likelihood obtained in the previous step. The *Q*-function is the core of the EM algorithm. It refers to the expectation of the log-likelihood function for complete data with respect to the conditional probability distribution over unobserved data given the observed data and current parameters [45].

If the posterior probability is used to represent the correspondence between *X* and *Y*, then
(13)$$p(u|x_{v})=\frac{P(u)p(x_{v}|u)}{p(x_{v})}$$

According to the expectation step in the EM algorithm, the *Q*-function can be obtained:(14)$$Q=\sum_{v=1}^{V}\sum_{u=1}^{U+1}P^{old}(u|x_{v})\log(P^{new}(u)P^{new}(x_{v}|u))$$
where $P^{old}$ represents the posterior probability calculated from the prior parameter value $P^{new}$. To simplify the *Q*-function, we remove the constants that are not related to $\left(\theta ,\sigma^{2}\right)$, and Equation (14) is rewritten as follows:(15)$$\left\{\begin{array}{c} Q(\theta ,\sigma^{2})=\frac{1}{2\sigma^{2}}\sum_{v=1}^{V}\sum_{u=1}^{U}P^{old}(u|x_{v}){\left\|x_{v}-T(y_{u},\theta )\right\|}^{2}+\log\sigma^{2}\sum_{v=1}^{V}\sum_{u=1}^{U}P^{old}(u|x_{v})\\ P^{old}(u|x_{v})=\frac{\exp\left(-\frac{1}{2}{\left\|\frac{x_{v}-T(y_{u},\theta^{old})}{\sigma^{old}}\right\|}^{2}\right)}{\sum_{u=1}^{U}\exp\left(-\frac{1}{2}{\left\|\frac{x_{v}-T(y_{u},\theta^{old})}{\sigma^{old}}\right\|}^{2}\right)+2\pi {\left(\sigma^{old}\right)}^{2}\frac{q}{1-q}\frac{U}{V}} \end{array}\right.$$

Next, we define $T(y_{u},\theta )=T(y_{u},D,t)$ as the affine variation between point clouds of two cameras, where *D* is the matrix of affine variation and *t* is the translation vector. Let $P=P^{old}(u|x_{v})$, and then Equation (15) changes to the following:(16)$$Q(D,t,\sigma^{2})=\frac{1}{2\sigma^{2}}\sum_{v=1}^{V}\sum_{u=1}^{U}P{\left\|x_{v}-(Dy_{u}+t)\right\|}^{2}+\log\sigma^{2}\sum_{v=1}^{V}\sum_{u=1}^{U}P$$

*D*, *t*, and $\sigma^{2}$ in the maximize *Q* can be solved by directly making the partial derivative of *Q* equal to zero. These values can be used to update *P*. These steps are repeated until convergence is reached. Then, we can obtain the aligned point clouds as follows:(17)$$X=T(Y,D,t)=YD^{T}+1t^{T}$$

In this process, *P* provides probabilistic correspondences. Each point in the left (or right) camera (point clouds *Y*) will be matched with the point in the right (or left) camera (point clouds *X*). These points have the highest probability of correspondence in the whole points cloud, which is selected from the number of *V*.

## 3. Results

### 3.1. Design of 7 × 7 DOE with Uniform Energy Distribution

Based on the above theory, we designed and processed a DOE that can divide incident light into 7 × 7 output beams. Its maximum diffraction angle is 35° and the overall diffraction efficiency is about 79%. The operating wavelength is 940 nm. Silicon with a thickness of 0.181 μm and a refractive index of 3.60 was chosen as the structural material. Sapphire with a refractive index of 1.76 was chosen as the substrate material. According to the grating equation, the period of the unit cell of the DOE is 6.953 μm × 6.953 μm. The incident light polarization was set along the *Y*-axis in Figure 2B. The optimization process stops when the *EF* is less than 0.02 or the number of iterations exceeds 100.

Figure 2 demonstrates the optimization process of the 7 × 7 DOE. The *EF* changes from 0.379 to 0.207, and the diffraction efficiencies at all orders of the initial and final solutions are presented as bar charts in Figure 2A, which shows that the diffraction efficiency of the DOE becomes more uniform during the optimization process. The phase distribution of the initial and final solutions is shown in Figure 2B (upper panels) and the distribution of the phase difference is thus obtained, as revealed in Figure 2B (lower panel), emphasizing the phase difference between these two solutions.

For the DOE processing, we use the method of photolithography, by etching the substrate to form a notch to obtain the DOE. The detailed process is as follows: mask fabrication, coating, pre-baking, exposure, development, film stabilization, etching, and degumming cleaning. First, based on the simulation results, the mask pattern for photolithography is designed, and the diffraction amplitude mask is generated by the pattern generator. Next, the photoresist is coated on clean sapphire, and the coating should be uniform without undulations or pinholes. After this step, the sapphire is placed in a thermostatic container to give a certain time of hot baking to increase the adhesion and the abrasion resistance of the adhesive film. Place the photolithography mask plate on top of the sapphire, align the graphics on the mask plate with the sapphire position, and expose it under the light source to control the exposure time. The exposed sapphire is put into the developing solution to wash away the light-sensitive part of the photoresist and keep the un-sensitive photoresist. The mask graphics will be transferred to the photoresist. Then, it is time to check whether the graphics are clear and correct. Developing will make the adhesive film expand and soften, so the adhesive film should be dried to remove moisture to enhance the adhesive film resistance after being developed. Then, the sapphire is ready to etch. The part retaining the photoresist on the sapphire is not etched, while the exposed part is etched to a certain depth. Finally, we clean the remained photoresist on the sapphire after etching and we obtain the two-level DOE. Figure 2C shows the electron microscope scanning images of the 7 × 7 DOE.

### 3.2. Structured-Light 3D Imaging and Target Reconstruction

A binocular vision system is a system that uses two cameras or camcorders to mimic the principle of stereoscopic vision in human eyes, enabling more accurate and reliable 3D reconstruction. Each camera captures a different viewpoint of the same scene. By analyzing the difference between the two viewpoints, information about the depth, distance, and shape of the scene can be obtained. The reason for choosing binocular vision systems in structured-light 3D imaging is that they can provide additional information of depth, increasing the accuracy and precision of 3D reconstruction, as well as possessing strong anti-interference capabilities. On the other hand, the employed VCSEL is a type of laser that forms an optical resonance cavity perpendicular to the semiconductor epitaxial wafer and emits a laser beam perpendicular to the surface of the substrate, which has the advantages of small size, low power consumption, high efficiency, long lifetime, and two-dimensional surface-array integration. It is suitable for providing point light sources for structured-light 3D imaging. The system is built up as schematically depicted in Figure 3A, where the DOE and VCSEL are assembled and placed between the binocular camera, used as an illumination source. The binocular camera collects the images of the target object separately. The objects to be measured in this experiment are a full-face mask placed in front of the system and a white paper plane under the mask with different depths. This system projects a field of view up to 35° × 35°, with an effective number of 8869 projected points (VCSEL produces 181 point light sources).

In this system, once the parallax values of all the effective pixels in the image captured by the camera are accurately obtained, the depth calculation can be performed by the triangulation principle, which is based on the binocular vision system as shown in Figure 3B. The acquisition of depth is obtained by the following Equation (18):(18)$$Z=\frac{fB}{x_{1}-x_{2}}$$
where Z is the object’s depth; $x_{1}$ and $x_{2}$ are the coordinate points on the image planes of the left and right cameras, respectively, both of which correspond to the same point on the real image; f is the focal length of the camera, and B is the distance between the two cameras. From the Equation (18), the depth Z is inversely proportional to $x_{1}-x_{2}$. After extracting the set of point locations in the images captured by the left and right cameras, the two sets of point locations are subjected to point matching, which is one of the key steps in performing 3D reconstruction. After precisely determining the location of each point on the surface of the target object from different viewpoints, the parallax between the left and right cameras (i.e., $x_{1}-x_{2}$ in Equation (18)) can be obtained. By applying Equation (18) with the pre-known values of f, B and the deduced value of $x_{1}-x_{2}$, the depth information at each point is then obtained and the 3D reconstruction of the target image is thus eventually realized.

The reconstruction algorithm of this system adopts the CPD algorithm [43], the main idea of which is to consider a point set as a sample generated by a probability density function, and then describe this probability density function with a GMM to realize the correspondence point matching. In the CPD algorithm, the correspondence between the source and target point clouds is realized by calculating the least squares solution between two GMMs. Compared with the traditional matching methods, the CPD algorithm has higher matching accuracy and robustness to outliers and missing points. The recovery results of this system are shown in Figure 3C. The depth of the full-face mask is clearly shown. The white paper plane with a depth difference with the mask can also be seen in the top view. It aims to show the difference in depth between the two objects.

## 4. Discussion

Vector IFTA improves the efficiency of DOE design and processing. It adds a vector electromagnetic simulation part to the scalar diffraction theory, realizing a more uniform diffractive device. However, there is still room for further optimization of this algorithm. There is a possibility for adjusting the selection of *m* in iterations or the strength of the correction function. Selecting the metasurface with a smaller unit cell period is another way to improve the efficiency of this algorithm. This metasurface allows more precise modulation of the incident electromagnetic wave, which results in a larger field of view and higher diffraction efficiency compared to conventional DOE.

It can be seen that the diffraction efficiency of the DOE is not completely uniform. There is higher diffraction efficiency in the (5, 5) part in Figure 2A, which is related to the not-perfect optimization. Interestingly, in the actual DOE design, higher diffraction efficiency will happen in the center order from the design due to the etching process. The larger the deviation is, the more the energy of the center order will increase [46]. In addition, the increase in the number of beams and diffraction angle increases the optimization difficulty.

Based on Figure 3C, it can be seen that the overall contour of the full-face mask was successfully reproduced. However, there is a small part of the mask missing near the left cheek, which is only related to the placement and angles between the cameras and the object. In this system, the full-face mask and the white paper plane are located at an angle of about 15 degrees to the camera. There are about 8000+ light spots on the 50 cm × 50 cm object, and the accuracy of the reconstruction is less than 0.3 cm^2^ per point.

In addition, it should be noted that we first calibrated both the internal and external parameters of the camera before the experiment, which ensures the high accuracy and stability of the whole system and also improves the accuracy and reliability of the 3D reconstruction [47]. In this paper, we obtain the internal and external parameters of the binocular vision system with the help of Stereo Camera Calibrator, which is a toolbox of MATLAB R 2022b. This toolbox adopts the stereo vision model based on the principle of triangulation. The calibration results are shown in Table 1.

The combination of IFTA and structured-light 3D imaging technology accelerates the development of 3D imaging technology, and this work can be applied in virtual reality, face recognition, automatic driving, and other intelligent fields. It creates a new method for light control and provides a new path for imaging.

## 5. Conclusions

In summary, in this paper, a quasi-continuous-phase metasurface is designed and processed based on Vector IFTA and imaged as a 7 × 7 DOE in structured-light 3D imaging. This quasi-continuous-phase metasurface is combined with VCSEL to generate point light sources, which are illuminated on a full-face mask and a white paper plane with different depths. Image acquisition is performed by a binocular vision system to complete the imaging and reconstruction of the 3D objects. The maximum diffraction angle of the DOE can reach 35°, and the overall diffraction efficiency is close to 79%. The iteration speed of the algorithm is rapid and the reconstruction effect of the target object is accurate. This method provides a new technical path for 3D imaging and has a broad application prospect.

## Figures and Tables

**Figure 1 nanomaterials-14-00929-f001:**
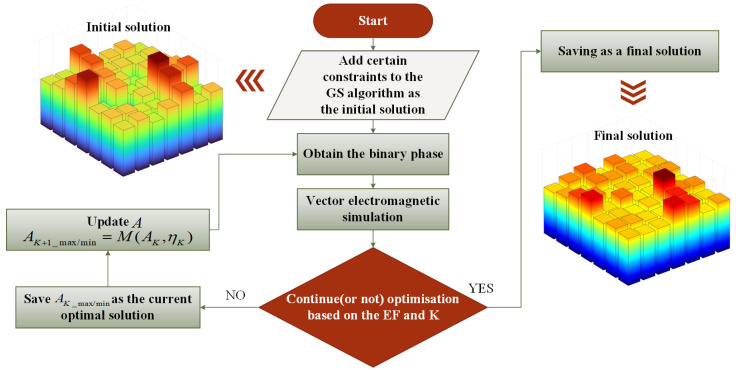
Optimization flow of Vector IFTA. First, the initial solution is obtained by the revised GS algorithm, and vector electromagnetic simulation is performed on it. Then, optimized results are judged according to the evaluation function and the number of iterations. Finally, the quasi-continuous-phase metasurface with diffraction uniform is obtained.

**Figure 2 nanomaterials-14-00929-f002:**
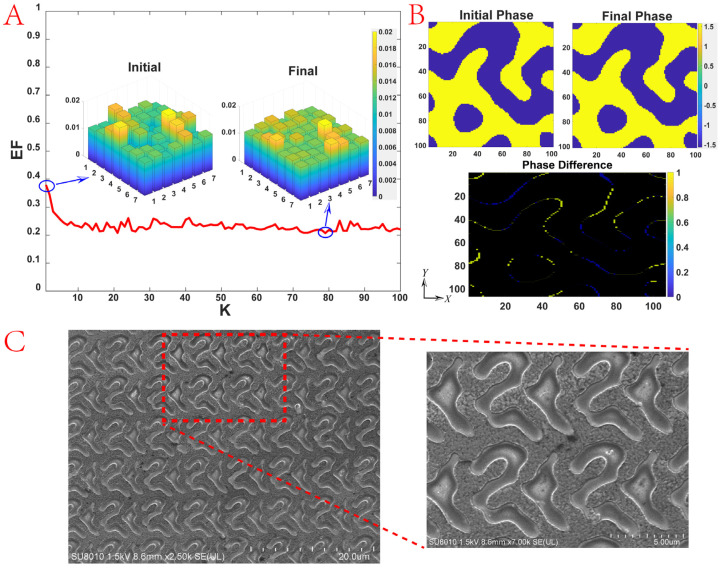
The optimization process of the 7 × 7 DOE. (**A**) Evaluation function *EF* versus the number of iterations K. The diffraction efficiencies at all orders of the initial and final solutions are also shown. (**B**) Phase distributions of the initial solution, final solution, and the difference between them. (**C**) The electron microscope scanning images of the 7 × 7 DOE.

**Figure 3 nanomaterials-14-00929-f003:**
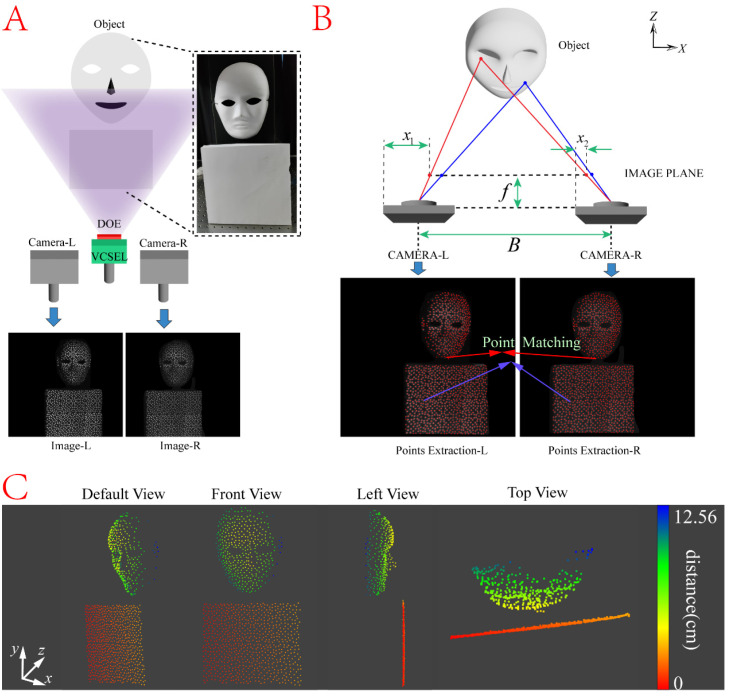
Principles and results of structured-light 3D imaging and target reconstruction. (**A**) Setup of the binocular vision system. DOE and the VSCEL are integrated together and placed between the two cameras. The target objects are a full-face mask and a white paper plane with different depths and heights. (**B**) Principle of acquiring depth information and point matching for the binocular vision system. (**C**) Result of structured-light 3D reconstruction with different viewing directions.

**Table 1 nanomaterials-14-00929-t001:** The calibration results of the binocular vision system.

	Parameter	Result
Camera—Left	Internal Reference Matrix	$\left[\begin{array}{ccc} 2870.553 & -0.061 & 1269.105\\ 0 & 2871.493 & 941.98\\ 0 & 0 & 1 \end{array}\right]$
	Distortion Factor	−0.197, 0.183, −0.156, −0.00007, 0.00065
Camera—Right	Internal Reference Matrix	$\left[\begin{array}{ccc} 2869.618 & -0.097 & 1278.367\\ 0 & 2869.667 & 949.371\\ 0 & 0 & 1 \end{array}\right]$
	Distortion Factor	−0.1897, 0.093, 0.1385, −0.00001, −0.0002
System	Rotation Matrix	$\left[\begin{array}{ccc} 1 & 0.001 & -0.003\\ -0.0005 & 1 & 0.002\\ 0.003 & -0.002 & 1 \end{array}\right]$
	Translation Matrix	$\left[\begin{array}{ccc} 105.76 & -0.0041 & 0.9302 \end{array}\right]$

## Data Availability

The data presented in this study are available in the article.

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
