# Peer review of "Structured-Light 3D Imaging Based on Vector Iterative Fourier Transform Algorithm"

_nanomaterials, 2024, doi:10.3390/nano14110929_

Round 1

Reviewer 1 Report

Comments and Suggestions for Authors

Manuscript ID_nanomaterials-2988792

Reviwer’s comments:

1. The authors designed a quasi-continuous-phase metasurface beam splitter through a vector iterative Fourier transform algorithm. The device was used to realize structured light 3D imaging of a target object and reconstruction.

2. vertical-cavity surface-emitting laser (VCSEL) In the explanation below Fig. 3 , and in the Conclusions: instead of VSCEL should be VCSEL.

3. r.129, after “evaluation function”, please write “EF”.

4. More clear information about the experimental part are needed and would improve the paper. It is not clear how the DOE is obtained (section 3). A photo of the DOE and of the entire experimental set-up would help the reader to understand.

5. In the Introduction section, the author might also refer to the recently published papers:

DOI: 10.3390/ma14092201 and: DOI: 10.3390/ma14247842

The paper can be published after the above mentioned issues are addressed.

Reviewer 2 Report

Comments and Suggestions for Authors

The article is interesting, but requires some changes.

What is metasruface?

What do we mean by structured light?

Figure 1 is very small, especially the font used. Similarly, Figure 2. It seems that good drawings constitute the quality of an article, especially if it is about imaging. Considering such a topic, the graphic aspect of the article is very poor. In turn, the font in table 1 is very large.

What is the purpose of this article? A combination of 3 elements: imaging method, algorithm and measurement?

What's new in this article? Metasurface, algorithm, measurements, reconstruction?

The constraints for the vector iterative FT algorithm are already used in the literature, are the ones used here different?

Line 100 uses the term singularity suppress. What does this mean and what is its negative impact on reconstruction?

The issue of the quality of reconstruction has not been fully explained in the research conducted. What is the accuracy of the reconstruction?

Were measurements only made on one geometric shape?

Authors should refer particularly carefully to their previous work:

https://www.mdpi.com/1996-1944/14/4/1022

Quasi-Continuous Metasurface Beam Splitters Enabled by Vector Iterative Fourier Transform Algorithm

But also, for similar works:

Zou, X., Lin, R., Fu, Y., Gong, G., Zhou, X., Wang, S., ... & Wang, Z. (2024). Advanced optical imaging based on metasurfaces. Advanced Optical Materials12(6), 2203149.

Pryor Jr, A., Yang, Y., Rana, A., Gallagher-Jones, M., Zhou, J., Lo, Y. H., ... & Miao, J. (2017). GENFIRE: A generalized Fourier iterative reconstruction algorithm for high-resolution 3D imaging. Scientific reports, 7(1), 10409.

Round 2

Reviewer 1 Report

Comments and Suggestions for Authors

The paper can be published now.

Reviewer 2 Report

Comments and Suggestions for Authors

The authors answered my questions very carefully and made appropriate changes to the article. The article may be published in the current version.